# Recent Advances in Conventional Methods and Electrochemical Aptasensors for Mycotoxin Detection

**DOI:** 10.3390/foods10071437

**Published:** 2021-06-22

**Authors:** Jing Yi Ong, Andrew Pike, Ling Ling Tan

**Affiliations:** 1Southeast Asia Disaster Prevention Research Initiative (SEADPRI-UKM), Institute for Environment and Development (LESTARI), Universiti Kebangsaan Malaysia, UKM Bangi, Selangor Darul Ehsan 43600, Malaysia; ongjy75@gmail.com; 2School of Natural and Environmental Sciences, Bedson Building, Newcastle University, Newcastle upon Tyne NE1 7RU, UK; andrew.pike@ncl.ac.uk

**Keywords:** mycotoxin, electrochemical, aptasensor, biosensor

## Abstract

The presence of mycotoxins in foodstuffs and feedstuffs is a serious concern for human health. The detection of mycotoxins is therefore necessary as a preventive action to avoid the harmful contamination of foodstuffs and animal feed. In comparison with the considerable expense of treating contaminated foodstuffs, early detection is a cost-effective way to ensure food safety. The high affinity of bio-recognition molecules to mycotoxins has led to the development of affinity columns for sample pre-treatment and the development of biosensors for the quantitative analysis of mycotoxins. Aptamers are a very attractive class of biological receptors that are currently in great demand for the development of new biosensors. In this review, the improvement in the materials and methodology, and the working principles and performance of both conventional and recently developed methods are discussed. The key features and applications of the fundamental recognition elements, such as antibodies and aptamers are addressed. Recent advances in aptasensors that are based on different electrochemical (EC) transducers are reviewed in detail, especially from the perspective of the diagnostic mechanism; in addition, a brief introduction of some commercially available mycotoxin detection kits is provided.

## 1. Introduction

Mycotoxins have been an agricultural concern ever since the emergence of cultivation activities when humans first started to farm and store their food [1]. A mycotoxin is a secondary metabolite, mainly produced by the *Fusarium* and *Aspergillus* fungi that are found on foodstuffs and feedstuffs [2]. The first mycotoxin, aflatoxin (AF), was discovered after a severe incident where 100,000 turkeys died after consuming contaminated groundnuts in 1962 [3]. Since then, many types of mycotoxins such as citrinin (CIT), fumonisin (FUM), ochratoxin A (OTA), patulin (PAT), zearalenone (ZEN), ergot alkaloids and trichothecenes such as T-2 toxin (T-2) were characterised during the 1960s and 1970s [3]. Some of the chemical structures of common mycotoxins in foodstuffs and feedstuffs are shown in Figure 1 [4,5,6].

At present, mycotoxins are still a global concern, as they are prevalent in foodstuffs and feedstuffs that are widely consumed, such as maize, wheat, peanuts, milk and sorghum. Researchers have found that the maize and wheat in China, Africa and Europe are contaminated by various mycotoxins to different degrees [7,8]. The prevalence of mycotoxin types differs according to the environmental conditions. In China, contamination with aflatoxin B_1_ (AFB_1_), deoxynivalenol (DON) and ZEN is prevalent in maize and wheat, whereas FUM and AFB_1_ are prevalent in maize and sorghum grain, respectively, in Africa [7,8]. The situation in Europe is estimated to worsen in the next 30 years due to climate change, which leads to more favourable conditions for *Aspergillus flavus* to proliferate [9]. The production of AFs, CIT, OTA and PAT is often caused by the *Aspergillus* species, while ZEN and DON are produced by the *Fusarium* species [6]. Mycotoxin contamination of foodstuffs and feedstuffs is a potential threat to public health and is especially concerning in China, one of the main global distributors of maize and wheat [10]. Therefore, effective and affordable methods for mycotoxin detection are indispensable for maintaining high levels of global food safety. Recently, the classification, toxicity, characteristics and contamination of mycotoxin in specific food and feeds were reviewed along with several analytical methods, including instrumental and sensor techniques [5,6,11]. Biosensors with different recognition elements that focus on nanostructured materials and analytical techniques such as electrochemistry, electrochemiluminescence and photoelectrochemistry are the recent trend in mycotoxin detection [12,13,14].

Herein, the current trends in mycotoxin detection methods, including both conventional and more recently developed methods, are discussed in detail. The current preferred method is based on an electrochemical (EC)-based aptasensor. Different types of molecular recognition elements that are used in the development of biosensors, such as antibodies, aptamers (Apts) and molecularly imprinted polymers (MIPs), are reviewed. Furthermore, the key commercially available mycotoxin detection products are discussed in order to provide an overview of the technologies that are in current use. This review is intended to pave a clear path for the future advancement of mycotoxin detection.

## 2. Conventional and Advanced Analytical Technologies

The detection of mycotoxins has been explored for nearly 60 years, ever since their characterisation in the early 1960s. Although around 400 mycotoxins have now been discovered, only a portion have high toxicity and are also prominent in foodstuffs and feedstuffs [2]. Screening for mycotoxins is the most cost-effective way to prevent harmful mycotoxins from entering the food chain. Therefore, many analytical techniques have been developed to ensure food safety and security.

Most of the analytical techniques are based on chromatographic and immunological methods. Large instruments, such as high-performance liquid chromatography (HPLC), liquid chromatography–mass spectrometry (LC-MS) and gas chromatography–mass spectrometry (GC-MS), were routinely used during the early developmental stages of mycotoxin detection. Their key advantages are that they offer good selectivity, high sensitivity and also a low limit of detection (LOD) whilst providing high throughput [15]. Thus, these chromatography methods are now used as a reference point for alternative mycotoxin detection methods that are now being developed [16]. However, these conventional methods are extremely expensive and require trained personnel to operate. Alternatively, the enzyme-linked immunosorbent assay (ELISA) approach offers a convenient route to filter out any contaminated samples from within a large batch of samples. Nevertheless, in ELISA, the possibility of false positives is high due to both the matrix effect and the fact that the antibodies are often highly unstable [17,18].

Over the years, various types of sensors, including biosensors, EC sensors and optical sensors, have been developed as alternatives for the established conventional methods. Some recent enhancements to the conventional molecular-based recognition methods that are fundamental to the detection of mycotoxins are discussed in the following section, and the significance of different types of aptasensors, including electrochemical aptasensors, is also reviewed.

### 2.1. Molecular Recognition Elements

#### 2.1.1. Antibodies

Molecular recognition elements enable a sensor to function by specifically binding the desired analyte, resulting in changes that can be measured and analysed by the transducer of the biosensor. This means that the sensitivity of the biosensor is directly attributed to the performance of the molecular recognition elements [19].

An antibody is one such conventional molecular recognition element, and is a well-established component in analyte detection due to the specific antibody–antigen interaction. It comprises a fragment crystallisable (Fc) region and an antigen-binding fragment (Fab) region that are made up of two heavy chains (H) and two light chains (L), as illustrated in Figure 2a [20,21].

Antibodies for a species of interest are obtained from an animal which has been immunised with the particular antigen. The antibodies in antisera are then extracted and their affinity towards the antigen is evaluated by ELISA techniques. Although polyclonal antibodies can be produced in a shorter timeframe, the binding sites of the polyclonal antibodies to the antigen are different, since each clonal antibody has an affinity to different epitopes of the same antigen [22]. Thus, monoclonal antibodies (mAbs) with an affinity to the same epitopes of the antigen are strongly preferred in therapeutic applications and immunotherapy [23,24,25].

Studies involving antibodies in the detection of mycotoxins are tabulated in Table 1. mAbs are mainly produced by hybridoma technology, which was invented in 1975 [26]. The inherent pairing information of the resultant antibody is maintained, providing high specificity and sensitivity to the targeted antigen [26]. mAbs are also used in the application of AFB_1_ detection by using fluorescence immunoassays (FLISAs) [27]. Broad-specific mAbs were developed for the detection of total aflatoxins (AFs) (AFB_1_, aflatoxin B_2_ (AFB_2_), aflatoxin G_1_ (AFG_1_) and aflatoxin G_2_ (AFG_2_)), with a low half maximal inhibitory concentration (IC_50_) in the range of 0.04 µg kg^−1^ to 0.06 µg kg^−1^. This was achieved by incorporating an AFB_1_ modified with O-carboxymethyl oxime (AFB_1_-CMO), which is known to produce broad-specific mAbs, as the hapten in antibody production, followed by cell fusion with hybridoma technology [28]. There are other commercial monoclonal antibody isolation techniques that are available, and they vary depending on the intended application of the antibodies. For instance, Sino Biological Inc. located in Beijing, China provides options such as proprietary rabbit monoclonal antibody technology, single B cell technology, fast mouse monoclonal antibody and phage display technology for the production of mAbs.

The disulphide bridge in an antibody can be cleaved by using a solution of Tris(2-carboxyethyl) phosphine hydrochloride (TCEP), resulting in two antibody fragments, as shown in Figure 2b. An EC immunosensor for amyloid-β fibril detection was developed by immobilising one half-antibody fragment onto a polycrystalline gold electrode surface via a covalent bond between the cleaved sulphide and a gold particle [29]. The antibody can also be further fragmented into a single-chain antibody fragment (scAb), as shown in Figure 2d. It is formed from the bonding of a single-chain fragment variable (scFv), consisting of variable domains known as the variable light (V_L_) and variable heavy (V_H_) domains, as illustrated in Figure 2c, to the constant region at the terminal V_L_ or V_H_ of the scFv. In recent years, scAbs have been widely utilised as sensing elements due to their robust characteristics. A scAb (8 nm) is around half the size of a complete antibody (14 nm), thus providing the potential for higher surface density [30]. In addition, a scAb allows for flexibility in terms of customisation and immobilisation, while also providing better sensitivity and specificity in the formation of immune complexes [31]. Nevertheless, a scFv can also work as a sensing material with hybridoma technology without being further converted into a scAb. Some examples are the detection of fumonisin B_1_ in maize and malachite green in tilapia fish [32,33].

At present, antibodies are still widely used and are a powerful tool in the development of recognition elements due to the reasons mentioned above. Even Apt originated from part of an antibody, and its key features are discussed in the following section.

#### 2.1.2. Aptamers

Apts are categorised into two categories: oligonucleotides and peptide molecules. Oligonucleotide-based Apts capture target molecules via complementary shapes and electrostatic and hydrophobic interactions, whereas peptide Apts consist mainly of variable peptide domains that bind to a protein scaffold at both terminal ends [18,34,35]. The term “aptamer” was derived from the word “*aptus*” in Latin, which means “to fit” and it was used to name the nucleic acid-based (RNA) ligands that were selected with the newly developed systematic evolution of ligands by exponential enrichment (SELEX) technique in 1990 [34].

SELEX was independently developed by two laboratories: the Gold lab and the Szostak lab [34]. The conventional SELEX technique begins with the selection of oligonucleotide sequences that have a high affinity for the targeted compound from a library of oligonucleotides with different sequences. Next, the unbound oligonucleotides are removed, while the oligonucleotides bound to the target compounds are isolated and amplified by polymerase chain reaction (PCR) or reverse transcription PCR (RT-PCR) to obtain the Apt sequences. The procedure is repeated several times in order to obtain Apt sequences that are highly specific to the targeted compound [36].

Similar to antibodies, Apts are widely used in the pharmaceutical sector for therapeutic purposes, such as in the targeted transport of therapeutic agents, and cancer diagnosis and therapy [37,38]. The popularity of Apts over antibodies has resulted from their higher effectiveness in delivering anticancer medicine into tumorous cells due to their small molecular size, which facilitates cell penetration [39]. Furthermore, Apts offer other benefits, including ease of production, stability, facile modification and immunogenicity in addition to their small size, which have been summarised in recent reviews [39,40].

Since the discovery of Apt, more than 2000 types of Apt have been created by the SELEX method [41]. The application of Apt as a recognition element in detecting food contaminants has been growing as more Apt sequences have been developed. Currently, Apt sequences for the commonly occurring mycotoxins, such as AFB_1_, AFB_2_, AFM_1_, OTA, ZEN, FUM B_1_, PAT and T-2 toxins, have been successfully identified [42,43,44,45,46]. Apts are selected using a range of modified techniques that are based on the SELEX procedure. During the SELEX process, Apts that successfully bind target molecules have been isolated via magnetic beads, affinity columns and capillary electrophoresis (CE), amongst other methods [47,48,49].

The type of isolation method for Apts is important in order to produce an Apt with high affinity and specificity to the target molecule. Various isolation methods have been developed and subsequently improved. A comparison of SELEX techniques that have been developed since 2016 was summarised in a recent review [50]. Among the various isolation techniques, CE is often favoured due to its flexibility to perform additional roles in SELEX other than simply isolation, such as analysis of the target purity, ssDNA library quality and Apt–analyte interactions. Furthermore, it is a low-cost technique which requires nanolitre volumes of the sample solution and a small amount of reagent, and uses a homogeneous (free) solution during the Apt–analyte binding step, which greatly reduces the number of repetitive rounds of complex amplification from 8–15 down to 1–4 rounds [41].

However, no particular SELEX method is applicable to every Apt selection. Thus, the design of each SELEX method has to be modified for every different case. For instance, Cell-SELEX is preferred when the information of the target protein is unknown, even though an extended screening time from repetitive cycles is required [51]. The Apts selected by different SELEX methods will possess different strengths of bimolecular interaction with the analyte. Thus, the optimum SELEX method has to be selected based on the type of target molecule to produce an Apt with high affinity. The binding affinity is indicated by the equilibrium dissociation constant (K_D_). A comparison of the K_D_ values of a range of Apts for small molecule targets, including saxitoxin, okadaic acid, OTA and α-amanitin, was summarised by Wang et al. [52].

Furthermore, in order to improve the stability and nuclease resistance of an Apt, inclusion of 5-chloro-2′deoxyuridine (5CIU) and 2′-phosphate groups has been exploited [51]. Apts can also be restructured into a secondary structure such as a “G-quadruplex” (G4) [53]. However, G-quadruplexes can only be formed from guanine-rich sequences because at least four guanine nucleotides are required. The structure of a G-quadruplex can be stabilised through the incorporation of polyaromatic groups as non-covalent ligands [54]. For example, the guanine and thymine in the G4 thrombin binding Apt (TBA) can be modified or substituted in order to increase its melting temperature (T_m_), enzymatic stability and binding affinity. The types of modification which strengthen or weaken the performance of TBA have been attentively reviewed by Riccardi et al. [53]. Apts can also undergo modification due to the local environment and their application. During the detection of OTA in acidic conditions, a thiolated Apt with a sulfhydryl group is preferred over an Apt with an amino group, since the thiolated Apt shows higher stability and reproducibility [55]. The versatility of Apt was also demonstrated by grafting the redox tag methylene blue onto an Apt. A change in the conformation of an Apt during AFB_1_ detection yields a current response as a result of changes in the electron transfer at an electrode’s surface [56].

In recent years, the alternative molecular recognition element of Apts has become more prominent, as it offers various advantages, such as a controlled method of synthesis, improved stability, facile modification and high specificity. The comparison of antibody and Apt-based detection approaches has been of much interest and has previously been discussed by several research groups [57,58]. Apts have a great potential to overcome the shortcoming of antibodies, and further efforts are necessary to develop, validate and verify Apt for more targets.

#### 2.1.3. Molecularly Imprinted Polymers

A MIP is a chemically synthetic sorbent, which is produced from three basic components: a functional monomer, a template and a cross-linker. Fabrication is directed by the target template, which serves as the moulding for the reactive site of the MIP, resulting in high selectivity towards the targeted compound. Therefore, MIPs perform well as molecular recognition elements. Although heterogeneous binding sites may be present during the synthesis of a MIP, it has some advantages that compensate for this drawback. Some key advantages of using a MIP as a substrate for mycotoxin detection is that its chemical and physical stability allows it to perform well within a complex matrix, such as in an environmental matrix which consists of a lot of impurities [59]. Generally, MIPs are synthesised in an organic solvent, which makes them excellent for the detection of organic compounds. This is due to the compatible diffusion ability of the MIP organic matrix with the organic solvent. On the contrary, a protein MIP is unable to be produced with an organic solvent due to diffusion difficulties, low solubility and conformational alterations; thus, such MIPs can only be produced in aqueous matrices [60]. In an attempt to widen the range of possible applications, research has focussed on different types of basic MIP components and the polymerisation techniques, especially in order to synthesise a MIP for use in an aqueous environment [61].

Many target samples for mycotoxin detection are aqueous, including those for environmental, food and therapeutic diagnoses [59,62,63]. Free radical and non-free radical polymerisation methods were developed to synthesise aqueous MIPs that avoid the occurrence of heterogeneous binding sites [61]. Currently, even green synthetic techniques, including ionic liquids, supercritical carbon dioxide and ultrasound, have been used to synthesise MIPs without the need for organic solvents [64].

Using a MIP as a sensing element faces the common problems of false positivity, low affinity and low selectivity. However, nanoparticles have been shown to enhance the characteristic response of MIPs [65]. A surface plasmon resonance (SPR) sensor for AFB_1_ detection exhibited high selectivity and good performance by incorporating gold nanoparticles (AuNPs) into hydroxyethyl methacrylate during polymerisation. The resulting MIP achieved an extremely low LOD of 0.001 μg kg^−1^ [66]. Therefore, MIPs are also good candidates as sensing elements, as long as suitable materials and synthetic methods are used.

### 2.2. Conventional Methods

#### 2.2.1. High-Performance Liquid Chromatography (HPLC)

The most common mainstream conventional techniques which have been validated for analysing mycotoxins are the chromatographic methods, such as thin-layer chromatography (TLC) and HPLC [67,68,69]. HPLC analyses samples by the isolation and quantification of an analyte from the sample matrix using a high-pressure solvent system [70]. The usual ultraviolet (UV) detector is often replaced by other detector types in order to achieve improved sensitivity and specificity. Over the years, HPLC methods have been enhanced by incorporating advanced chromatography columns, sample enrichment techniques and improved detectors, with the aim of obtaining refined peak resolution and low matrix interference. The growth of HPLC methods for mycotoxin detection in foodstuffs and feedstuffs with reference to these improvements is summarised in Table 2 and discussed below.

The atomic mass of a typical mycotoxin is less than 1000 Da, and thus, tandem mass spectrometry (MS/MS) is commonly coupled to HPLC due to its sensitivity to small molecules with an atomic mass below 5000 Da [71,72]. Among the different ionisation methods that have been investigated, which include electrospray ionisation (ESI), matrix-assisted laser desorption ionisation (MALDI), electron ionisation (EI) and atmospheric pressure chemical ionisation (APCI), the combination of ESI and the selected reaction monitoring (SRM) mode are commonly used in mycotoxin detection to effectively remove the mobile phase from the target ions [67,72,73]. This is largely due to the fact that mycotoxins are easily ionised in the positive mode of ESI, giving good peak resolution [74].

In 2011, an HPLC-MS/MS method that was able to simultaneously analyse up to four types of mycotoxin—AFB_1_, AFB_2_, AFG_1_ and AFG_2_—purely on their different retention times was reported [75]. To this day, no author has reported multi-mycotoxin detection that exceeded four types of mycotoxin, except for the identification of 14 types of secondary metabolites of fungi, which include AFB_1_ and AFG_1_ [76]. Although MS/MS has several advantages, such as rapid sampling and high reproducibility, it is very expensive and requires experts to analyse the MS spectra. In this case, the coupling of HPLC with a different detector, such as a diode array detector (DAD) or a fluorescence detector (FLD), is an alternative to MS/MS, since each mycotoxin possesses an intrinsic fluorescence characteristic [77,78]. Table 2 illustrates that analysis by FLD provides comparable recovery with MS/MS, although a longer response time is required.

Most of the reported studies into mycotoxin detection have adopted the maximum levels (MLs) from the Commission Regulation (EC) No. 1881/2006 as a reference in developing the method to comply with international standards in food safety and security. The minimum ML listed by the EC is 0.025 μg kg^−1^ for aflatoxin M_1_ (AFM_1_) in milk [79]. Identifying a mycotoxin with low concentration levels remains a challenge even for more sensitive detection systems. In terms of the matrix effect, the purity of the sample is crucial for correctly analysing mycotoxins at very low concentrations. Therefore, any impurities which will interrupt the analyses and could lead to false positive or false negative results must be removed. Therefore, sample enrichment is highly recommended before proceeding with a quantitative HPLC measurement of the mycotoxin. According to Beltran and co-workers, the sensitivity of ultra-HPLC tandem mass spectrometry (UHPLC–MS/MS) has failed to reach the MLs of AFB_1_ and AFM_1_ without pre-concentrating an extract taken from baby food and milk [80]. This issue was resolved by performing trace enrichment using solid phase extraction (SPE) with an immunoaffinity column (IAC). The ionisation suppression resulting from the sample matrix was removed by immunoaffinity purification, resulting in a higher sensitivity during quantification of the mycotoxin.

The advantages of IAC of increased efficiency and detection sensitivity have led to the further development of an improved version of IAC that is able to isolate various types of mycotoxin at the same time. A multi-immunoaffinity column (mIAC) was developed by incorporating different antibodies of mycotoxins in the IAC. Zhang and co-workers combined several sets of gel affinity columns, which were made up of cyanogen bromide (CNBr)-activated sepharose 4B and specific mAbs for the respective mycotoxins, to develop a single mIAC for the multi-detection of mycotoxins [81]. The in-lab mIAC can also be used twice, eluting the bound mycotoxin with methanol each time, thus improving its cost-effectiveness. It is possible to reuse a commercial IAC by proper regeneration of the antibodies’ binding sites. After elution, any remaining impurities on the commercially available IAC-ToxinFast column are rinsed off with phosphate-buffered saline (PBS) and water. The antibodies in the IAC are regenerated by immersion in PBS overnight at 4 °C before the next use. The IAC can be reused more than nine times by applying this simple rinsing and storing procedure, thus ensuring an economical sample enrichment technique prior to HPLC analysis [82].

An affinity column which substitutes antibodies with Apts was reported by Zhao et al. for the detection of AFB_1_ [83]. It has been proven that sample enrichment using an Apt affinity column (AAC) enables better peak resolution compared with IAC. Although it is possible to capture AFB_2_, AFG_1_, AFG_2_ and OTA, which all have similar chemical structures to AFB_1_, high recovery was only obtained for AFB_1_ [83].

Recently, a novel monolithic column was developed for sample extraction by using Apts as an alternative to antibodies. The Apt-based monolithic column was developed by Chen and co-workers by binding Apts to polyhedral oligomeric silsesquioxane (POSS) in order to create a highly selective active site for OTA [84]. Unlike AAC, the Apt-based monolithic column is highly selective to OTA. The selectivity of OTA is reinforced by increasing the hydrophilicity of the monolith column with N,N-methylene-bisacrylamide (MBA) [85]. In this system, a MIP can be copolymerised with the Apt to avoid unintentional adsorption by utilising the electrostatic repulsion between MIP and other substances [86]. Furthermore, the Apt-based monolithic column offers a high reusability of more than 30 times whilst maintaining satisfactory recovery, thus making it cost-effective [85]. In addition, Apt-based monolithic columns are highly flexible, since they are able to detect other mycotoxins if the corresponding Apt is used. High recovery levels are obtained from monolithic columns that are specially designed for PAT determination in fruit juice samples, apples and apple products [45]. Monolithic columns are different from affinity columns, since they are directly incorporated into the HPLC system. After the sample is percolated, it is injected into the analytical column for chromatographic separation and detection. Therefore, manual transfer of the sample solution from the affinity column into the HPLC analytical column is not required.

Although the HPLC method can provide increased sensitivity with MS-MS capabilities, it can be very time-consuming and costly, and sample enrichment techniques are exploited to allow the inclusion of cheaper detectors, such as DAD and FLD. Beyond the improvements in HPLC detectors, HPLC methods have been continually improved by the inclusion of enhanced sample enrichment and purification methods. Amongst these approaches, Apt-based extraction columns are currently favoured, since they offer excellent specificity and sample recovery.

#### 2.2.2. Gas Chromatography–Mass Spectrometry (GC-MS)

GC-MS became popular for mycotoxin detection during the 1970s after the successful analysis of AFs with TLC-MS [87]. Although the mycotoxin extraction process is similar to that for HPLC, the samples for GC-MS often require an extra step for analyte modification. Derivatisation is required to improve the volatility of the mycotoxins, particularly when they are present in trace amounts [88]. Two types of derivatisation are commonly used for mycotoxin detection: silylation and acetylation. Both methods substitute active hydrogen atoms in order to remove the strong polar hydrogen bond, which decreases the volatility of the analytes [89]. Ferreira and co-workers reported the silylation of DON and ZEN with trimethylsilyl derivatives, such as (N,O-bis(trimethylsilyl) acetamide) (BSA), trimethylchlorosilane (TMCS) and N-trimethylsilyimidazole (TMSI), observing recoveries in the range of 61–118% and 65–89% for unpopped popcorn and popped popcorn, respectively [90].

There have been cases where mycotoxin analyses by GC-MS were not reproducible due to inconsistent recovery. According to McMaster et al., even though the standard GC-MS operating procedure from the United States Wheat and Barley Scab Initiative (USWBSI) was adopted, the detection of DON in sorghum varied during the recovery process [91]. This issue was solved by diluting the sample with an internal standard, isotope d1-DON, during the sample preparation step [91].

To date, GC-MS methods have been limited to detecting only PAT and mycotoxins from *Fusarium* species such as ZEN, DON, T-2, HT-2 toxin (HT-2) and diacetoxyscirpenol (DAS) [92,93,94]. There are currently no reports on the detection of AFs and OTA, which are highly toxic to both humans and animals. Due to the extra derivatisation step and the high possibility of obtaining an inaccurate result, the GC-MS method is less favourable and has been less studied by researchers for mycotoxin detection.

#### 2.2.3. Enzyme-Linked Immunosorbent Assay (ELISA)

ELISA is a technique that exploits the specific binding between an antibody and an antigen, making it highly specific to the analyte. The different key characteristics of ELISA methods, such as direct, indirect, competitive and sandwich ELISA, have recently been reviewed by Sakamoto et al. [95]. Both qualitative and quantitative types of ELISA analyses depend on a colour change in the immunoassay due to a chromogen. Qualitative analyses are based on naked-eye observations of colour changes and intensity, whereas quantitative analyses require a microtiter plate reader which relies on either UV-Vis or fluorescence spectroscopy [95]. In the case of mycotoxin detection in foodstuffs and feedstuffs, such as peanuts, maize, nuts and milk, a direct competitive ELISA is commonly used. In competitive ELISA, the synthesised conjugated antigen and the antigen from the sample will compete to bind with the antibody prepared in the microtiter plate well. The difference between direct and indirect competitive ELISA is down to the type of antibody or antigen being labelled that is involved in the colour change [96]. For example, labelled antigens or primary antibodies are used in direct competitive ELISA, whilst indirect competitive ELISA incorporates labelled secondary antibodies as the reporters [96].

Qualitative detection of ELISA is preferable in mycotoxin detection for many samples, since high-throughput screening is also possible. Quantitative detection is not favoured because the high rate of false positive or false negative responses due to improper surface blocking during the analysis leads to inaccurate quantitation of mycotoxins [95,97]. The aim of monitoring food safety can be achieved without direct quantification, since the colour changes are directly correlated with the concentration of the mycotoxin. This can be attained on the premise that the concentration of antigen or antibody has been previously validated based on the maximum limits for food or feed safety. Recently, the application of the ELISA technique was enhanced by utilising different mechanisms for mycotoxin detection, as summarised in Table 3.

For instance, a colorimetric direct competitive ELISA utilises an indicator such as bromocresol purple (BCP) to represent the pH change upon enzymatic oxidisation of glucose by glucose oxidase (GOx) into hydrogen peroxide (H_2_O_2_) and gluconic acid [98]. Most of the ELISA techniques use horseradish peroxidase (HRP) to oxidise H_2_O_2_ due to its high affinity to H_2_O_2_ [99]. In one study by Xiong et al. [99], the yield of hydroxyl radical (**∙**OH) via oxidation of H_2_O_2_ by HRP was used to etch gold nanorods (AuNRs) into a smaller morphology. The colour of the AuNRs changed from bluish green to violet, followed by pink and orange as the **^∙^**OH concentration increased, thus allowing naked-eye detection of AFB_1_. The qualitative analysis of this study achieved a high sensitivity of 0.0125 μg kg^−1^ with a LOD of 0.004 μg kg^−1^ and a linear range of 0.0031 μg kg^−1^ to 0.1500 μg kg^−1^. On the other hand, the quantitative analysis provided an IC_50_ of 0.0223 μg kg^−1^, which is relatively low compared with a conventional ELISA (0.707 μg kg^−1^) that is based on a monochromic intensity change as a result of the enzymatic oxidation of 3,3′,5,5′-tetramethylbenzidine (TMB) by HRP [99].

Recently, an enhanced direct competitive ELISA was developed for the detection of AFB_1_ in corn with a higher sensitivity compared with colorimetric ELISA. The output signal was enhanced via dynamic light scattering (DLS). This study used a similar mechanism of oxidising glucose by GOx to produce H_2_O_2_, which was then oxidised into **^∙^**OH by HRP [100]. The sensitivity of this method was enhanced by amplifying the scattering signals from aggregations of the gold nanoparticles (AuNPs) induced by the formation of **^∙^**OH [93]. The signal amplification achieved a LOD (10% inhibitory concentration, IC_10_) of 0.00012 μg kg^−1^, with a regression equation of *y* = 16.899 ln(*x*) + 44.794 and an IC_50_ of 0.00136 μg kg^−1^, which are around 104- and 16-fold, respectively, lower than the aforementioned colorimetric ELISA [100].

ELISA is widely used as a preliminary analysis of the total AFs (AFB_1_, AFB_2_, AFG_1_ and AFG_2_) concentration. Subsequently, the exact concentration of each of the respective AFs can be determined and confirmed by HPLC methods simultaneously due to their different elution retention times [17,101]. Since ELISA is a reliable method for rapid screening, various commercial ELISA test kits have been developed [102]. However, further research to improve its performance and to lower the cost are being pursued. Some recent efforts have focused on improving the molecular recognition performance, as tabulated in Table 4.

For instance, seven mAbs possessing high sensitivity demonstrated the low IC_50_ values of 0.037 ± 0.002 µg kg^−1^ for AFB_1_ and 0.031 ± 0.001 µg kg^−1^ for total AF detection in peanut [103]. The mAbs for AFB_1_ can be replaced with nanobody immunomagnetic beads known as nanobody “Nb28”. A nanobody is a recombinant single-domain antibody engineered from a heavy-chain antibody to have various benefits, which surpass the normal complete antibody, such as being smaller (a diameter of only 2.5 nm and a height of 4 nm), improved solubility, stability and a higher resistance to denaturation while maintaining a comparable affinity and specificity to the normal complete antibody [104,105].

On the other hand, a mimotope is another alternative to chemosynthesised AFB_1_ conjugates that can also display high toxicity in a competitive ELISA. The term “mimotope” was formed from its action of mimicking the epitopes of a carbohydrate, protein or lipid antigen [106]. Therefore, a mimotope is able to compete with the target antigen in binding with the respective antibody. In this case, the mimotope was synthesised from peptide sequences that mimicked the epitope of nanobody Nb28. The structural gene for both nanobody Nb28 and its mimotope are known, thus reducing preparation costs [107].

Another application of a mimotope in the recent development of an immunosorbent assay has been reported by Peltomaa et al. [108]. A synthetic peptide mimotope for ZEN was determined with the DNA sequence of 5′-CCC TCA TAG TTT GGG TAA CG-3′ by phage display through consecutive selection of the targeted monoclonal antibody of ZEN [108]. It was then applied in a competitive up-conversion-linked immunosorbent assay (ULISA) through labelling with an optical label, which was constructed from a streptavidin (SA)-functionalised up-conversion nanoparticle (UCNP) via a poly(ethylene glycol) (PEG) linker [109]. This mimotope was able to achieve a LOD of 0.02 μg kg^−1^ due to the absence of any optical background effects by using UCNP-PEG-SA as a labelling agent.

Lately, a new nanozyme and Apt-based immunosorbent assay (NAISA) has been developed for AFB_1_ detection by altering the components involved in recognition, labelling and substrate absorption. Apts were used to capture AFB_1_ instead of the antibody, and the enzyme was substituted by a nanozyme composed of mesoporous silicon dioxide (SiO_2_), gold (Au) and platinum (Pt) nanoparticles. NAISA is highly selective and sensitive, achieving a LOD of 0.005 μg kg^−1^ [75].

The total time required to run a competitive ELISA, from the addition of mycotoxin to colour formation, is around 2 to 3 h, excluding the preparation steps of the 96-well microplates, which include coating overnight, blocking and the incubation of antibodies [98]. Although the response time is relatively long compared with HPLC methods, if analyses are done without duplication in a 96-well microplate, over 80 samples can be analysed at once, making the method attractive. Therefore, ELISA is an effective multi-sample rapid screening method for mycotoxins in foodstuffs and feedstuffs.

### 2.3. Electrochemical (EC) Aptasensors for Mycotoxin

The increased popularity of Apt has led to a similar increase in the development of Apt-based biosensors for mycotoxin detection [13,110]. A biosensor is composed of a bio-molecular recognition element and a transducer, which functions to convert the recognition activity into a measurable signal [111]. In this case, the Apt is the biomolecule, whilst the biosensor response can be obtained from a measurable electrical signal upon successful target binding by the aptamer.

The flexibility of Apts is reflected in the ease of modification of their 5′ and 3′ termini with functional groups like biotin, amines and thiols in order to bind in different ways to the modified surface of the biosensor. AuNPs are incorporated during surface modification for signal transduction or to increase the diagnostic sensitivity by enhancing the sensing surface area. Furthermore, the incorporation of conductive materials such as AuNPs enable direct application of Apt onto the electrode surface of an EC sensor [112,113]. Alternatively, the response from Apt can be indirectly detected with the aid of conductive substances that enable signal amplification. Recent developments in the area of EC aptasensors for mycotoxin will now be discussed.

Electrochemical impedance spectroscopy (EIS) is a sensitive EC technique for evaluating charge transfer processes, such as double-layer capacitance, impedance and the solution resistance within the interface between an electrode and an electrolyte [114]. EIS provides some advantages as a sensing technique; for example, the samples are not destroyed during analysis, and it is simple to operate while providing a large amount of information that is obtained by detailed analysis of the data [115]. For instance, the charge transfer resistance (R_CT_) of the Nyquist plot from EIS measurement is obtained from the Randles equivalent circuit that results from fitting the data [116]. Similarly, differential pulse voltammetry (DPV) and square wave voltammetry (SWV) are also simple to operate, and the mycotoxin concentration is detected by changes in the current as the potential is scanned over a suitable range.

EC methods that utilise different transducers, as shown in Table 5, focus on increasing the sensitivity of aptasensors via modification of the electrode surface, signal amplification or both. AuNPs are commonly coated onto the electrode surface to increase surface conductivity. It has been shown that the impedance of a boron-doped diamond (BDD) electrode decreased from 259 Ω to 166 Ω, showing improved electron transfer after AuNP modification [117]. Additionally, silver (Ag) metallisation is also effective. Ag^+^ ions have been coated onto the Apt after formation of an OTA–Apt complex and were then reduced to form Ag-coated Apt, which is electrically conductive, thus improving the yield signal [118].

Although using metal nanoparticles alone is effective, the rate of electron transfer can be further enhanced by enlarging the electrode surface area in order to bind more nanoparticles. The impedance of glassy carbon electrode (GCE) is lowered by the addition of iron based-metal–organic frameworks (MIL-101 (Fe)), which are octahedral in shape, thus binding more platinum nanoparticles (PtNPs) compared with a GCE modified with only PtNPs [119]. Carboxylated graphene is a cheaper alternative that is able to increase the electrode surface area while avoiingd the leaching of Apt [120]. Janus particles, which consist of different functional species in both of its hemispheres, were used as a bridge between a carboxylated graphene-modified GCE and an Apt. A single Janus particle is capable of binding to multiple Apts, whilst the carboxylated graphene provides many binding sites for the Janus particles. Thus, the synergic effect of both these materials enabled the highly sensitive detection of OTA with a LOD of 1.333 × 10^−6^ µg kg^−1^ and a linear range of 4.038 × 10^−6^–4.038 µg kg^−1^ [121].

Most of the aptasensors that use AuNPs for electrode surface modification exploit the use of a blocking agent, such as 6-mercapto-1-hexanol (MCH), in order to inhibit unintentional adsorption to exposed areas of the electrode surface. An extremely low LOD of 1.37 × 10^−6^ µg kg^−1^ was obtained with a wide linear range of 1 × 10^−5^ to 10 µg kg^−1^ for ZEN detection [122]. MCH is also used to ensure the linear structure of probe DNA strands in some aptasensors [123]. Aptasensors based on DPV often require a signal amplifier in addition to surface modification, as shown in Table 5. This is because the DPV current signal arises from an electrochemical reaction that is dependent on oxidation and reduction rates [124]. Most DPV aptasensors exhibit indirect detection of mycotoxin, whereby the oxidation or reduction is attributed to the competitive activity between the mycotoxin and a signal producing agent, as shown in Figure 3. For instance, Chen et al. [125] developed a DPV aptasensor by utilising the formation of an OTA–Apt complex that caused the OTA–Apt to dissociate from the modified electrode and to be replaced by a conductive conjugate, ferrocene-tagged AuNPs, as illustrated in Figure 4. Two AuNP conjugates were synthesised for electrode surface modification and signal amplification, respectively. The first AuNP conjugate was used for modification of the Au electrode surface. It was composed of a bridge probe (BP) that hybridised with the pre-coated capture probe (capture probe 1, CP1) on the Au electrode, an AuNP and another capture probe (capture probe 2, CP2), as shown in Figure 4a. Figure 4b depicts the second AuNP conjugate, which act as the signal amplifier in the assay. All the probes are thiolated DNA strands, which can bind to the Au surface via Au-S bonds. The competitive activity occurs between the OTA and the ferrocene-tagged AuNPs in the hybridisation with CP2, as shown in Figure 4c. The DNA duplex of CP2 and Apt will dissociate in the presence of OTA due to the higher affinity of Apt for OTA, thus enabling the hybridisation of CP2 and BP from the second AuNP conjugate. Therefore, the intensity of the cathodic current is directly proportional to the OTA concentration [125].

The signal-off behaviour is demonstrated by DPV aptasensors, where a smaller peak current is produced by a higher concentration of mycotoxin due to the changes on the modified electrode surface [126,127]. SA enhanced with an iron–porphyrin (PCN-223-Fe) composite can act both as a signal amplifier and also as a competitive agent, where it will bind to a 5′-biotin-modified Apt to produce a strong current peak upon oxygen reduction during the binding of SA and biotin in the absence of OTA. When OTA is present, the composite is replaced due to the higher affinity of the Apt for OTA, and thus no signal is produced. A good LOD of 1.4 × 10^−5^ µg kg^−1^ and a broad linear range of 2 × 10^−5^–2 µg kg^−1^ was obtained by using this mechanism [128]. A similar dual-target aptasensor was developed using two different complementary DNA (cDNA) sequences to the Apt for ZEN and FUM B_1_. Au, thionine (Thi) and 6-(ferrocenyl) hexanethiol (FC6S) were tagged onto the cDNA for signal amplification [128]. It was seen that without any surface modification or signal amplifier, the sensitivity of the aptasensor was greatly constrained; the LOD was very large (29.47 µg kg^−1^) compared with other aptasensors, as summarised in Table 5 [129].

The EC aptasensors that use a SWV technique are less frequently used in mycotoxin detection compared with EIS- and DPV-based sensors. A simple signal-on aptasensor was designed by Wang et al. [130] for AFB_1_ detection in alcoholic drinks, such as beer and white wine, as illustrated in Figure 5. In the presence of AFB_1_, the Apt of AFB_1_ will undergo dehybridisation with the complementary DNA (cDNA) of the Apt and will form a hairpin structure upon interaction with AFB_1_. The peak current in response to the concentration of AFB_1_ arises from the transfer of electrons between methylene blue (MB) at the 3′-end of the Apt and the Au electrode surface. Before formation of the hairpin Apt–AFB_1_ complex, the MB is positioned further away from the electrode surface, thus yielding a low peak current of MB. The peak current increases after formation of the Apt–AFB_1_ complex due to the conformation change of Apt, which shortens the distance between MB and the electrode surface. Therefore, the MB peak current is directly correlated to the AFB_1_ concentration, where the peak current increases with an increase in the AFB_1_ concentration [130]. Another aptasensor was similarly designed with the absence of cDNA and increased amount of labelled MB for detecting AFB_1_ in wine, milk and cornflour. MB was conjugated specifically onto the internal T site or 3′-end of the AFB_1_–Apt sequence instead of the 3′-end, as in the previous aptasensor. The specific MB-tagged Apt was in the form of a hairpin structure when it was immobilised on a gold electrode. The position of MB altered during the conformation change of the Apt upon formation of the Apt–AFB_1_ complex, and so facilitated the electron transfer between MB and the Au electrode. An aptasensor with a specific MB-tagged Apt achieved a LOD of 0.002 µg kg^−1^, which was lower than that of the system where the MB-tagged Apt was at the 3′-end (0.625 µg kg^−1^) [56]. Therefore, the design of an aptasensor that enables good electron transfer at the electrode surface is critical in obtaining a high-sensitivity response.

In summary, EC aptasensors are cost-effective and simple to fabricate, and possess high sensitivity. More DPV aptasensors are being developed due to their ability to produce sensitive signals by using a conductive signal amplifier, where the result is not affected by steric hindrance caused by a non-conductive Apt.

### 2.4. Commercial Mycotoxin Detection Kits

Commercial mycotoxin detection kits are available for both qualitative and quantitative analyses of foodstuffs and feedstuffs, such as peanut, corn, wheat and cottonseed. Most of the commercial products utilise the basic principles of ELISA in the formation of a coloured product after mycotoxin detection. A summary of the various commercially available mycotoxin detection kits for food and feed are tabulated in Table 6. The cost for each of the detection kits is not included, since it is not publicly available.

Many of the ELISA kits are not suitable for on-site detection, since they are designed to increase the efficiency of mycotoxin analysis in the form of a well-prepared 96 microtiter plates, which are immobilised with antibodies and all the necessary reagents, such as protein G and bovine serum albumin (BSA). Therefore, the analyses must be performed by trained personnel in a laboratory, since corrosive chemicals, such as concentrated hydrochloric acid (HCl), are often used as a stop solution for the chemical reaction. On the other hand, lateral flow immunoassays (LFI) are designed for on-site detection, as they rely on a simple operating procedure that supports both qualitative and quantitative analyses. Various recent developments in LFI that involve a range of different sensing agents, such as nanoparticles, enzymes, quantum dots, microspheres and other recognition elements for mycotoxin detection, have been discussed exhaustively in some recent reviews [131,132].

Most of the LFI summarised in Table 6 extract mycotoxin from food and feed by utilising methanol as the solvent and either an exerted vibration force that is generated by a vortex flow meter or manually by hand. The supernatant that contains the mycotoxin is then obtained by separation via centrifugation or gravity before it is applied to the lateral flow strip. Qualitative analyses are faster than quantitative analyses, since they do not require analysis of the lateral flow strip by a microplate reader. Both LFI and ELISA are able to comply with the maximum levels (<2 µg kg^−1^) regulated by the Commission Regulation (EC) Nos 1881/2006 and 1126/2007 for various mycotoxins, except for foods for babies and infants, which require a maximum level lower than 0.5 µg kg^−1^ [79,133]. Research has further improved the sensitivity of LFI by prolonging the reaction timeframe of the sensing mechanism and colour formation at the test line [134]. Recently, apart from microplate readers, the coupling of LFI with a smartphone-based application for quantitative analyses is becoming more commercially attractive due to the increased feasibility of a combination of techniques [135]. Therefore, the LFI method is suitable for on-site screening of mycotoxin, whereas ELISA is preferable for laboratory measurement of mycotoxin.

## 3. Conclusions

The comparison of all the technologies discussed in this review in terms of their advantages and disadvantages is summarised in Table 7. Currently, HPLC is widely used in the analytical laboratory for mycotoxin detection, since it is the recommended method in *Codex Alimentarius* for mycotoxin quantification [136]. In the HPLC analysis of mycotoxins, signal enhancement is observed in the ongoing development of new and improved types of detector, from DAD and FLD to MS/MS. In addition, Apt-based extraction columns have been developed to provide higher recovery of the target mycotoxin, which is now comparable with the affinity found in immunoaffinity columns. On the other hand, ELISA and lateral flow methods are commonly used for commercial detection of mycotoxins due to their robust ability in high-throughput screening and on-site detection, respectively.

The key current and most clearly apparent advancement in mycotoxin detection methodologies is the use of Apts instead of antibodies. This includes the use of Apts during sample extraction by an affinity column, their incorporation in the ELISA method (i.e., NAISA), and the development of EC biosensors. Apts are preferable due to several critical and promising properties: stability, flexibility, cost-effectiveness and efficiency in production. However, most of the current commercially available mycotoxin detection kits use antibodies as the recognition element. Thus, it is expected that the focus of future research will shift to the development of lateral flow assays that use Apts in order to lower the production costs whilst maintaining comparable performance. The SELEX method for the selection of the most appropriate Apts can also be enhanced by decreasing the screening time while still maintaining the binding affinity to target molecules. Different forms and sequences of Apt, which provide greater biological performance, such as stability, reproducibility, nuclease resistance and T_m_, are additional areas of research in the quest for an optimal EC aptasensor for mycotoxin detection.

## Figures and Tables

**Figure 1 foods-10-01437-f001:**
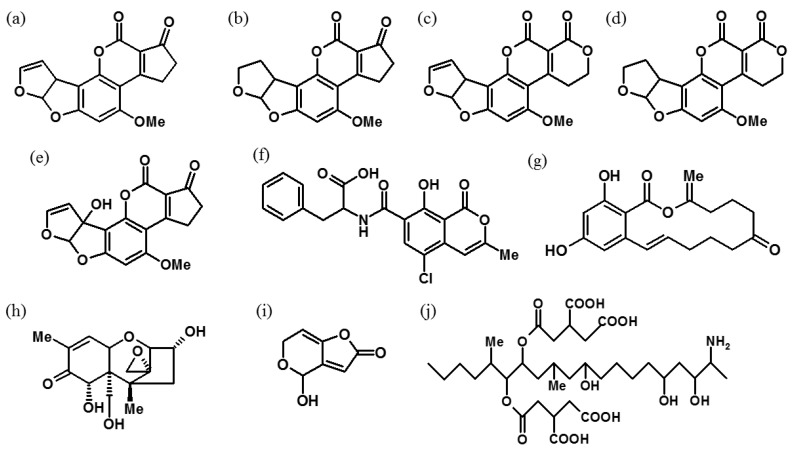
The chemical structures of (**a**) aflatoxin B_1_, (**b**) aflatoxin B_2_, (**c**) aflatoxin G_1_, (**d**) aflatoxin G_2_, (**e**) aflatoxin M_1_, (**f**) ochratoxin A, (**g**) zearalenone, (**h**) deoxynivalenol, (**i**) patulin and (**j**) fumonisin B_1_.

**Figure 2 foods-10-01437-f002:**
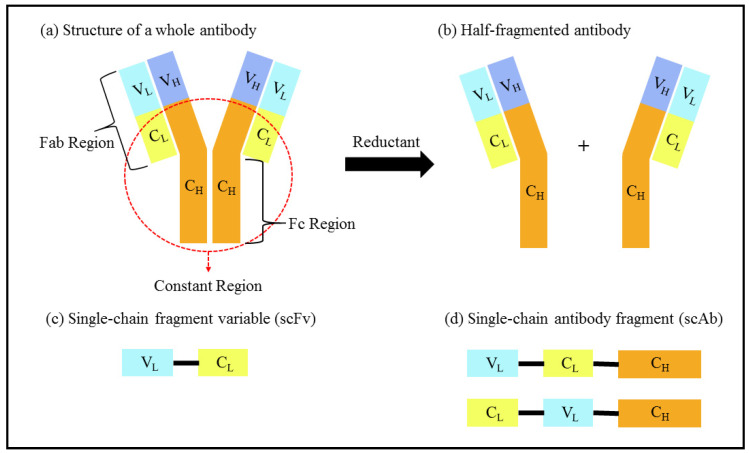
Schematic diagram of (**a**) the structure of a complete antibody and different types of antibody fragment after cleavage, (**b**) a half-fragmented antibody, (**c**) a single-chain fragment variable (scFv) and (**d**) a single-chain antibody fragment (scAb). V_L_, variable light; V_H_, variable heavy; C_L_, constant region in light chain; C_H_, constant region in heavy chain; Fab, antigen-binding fragment; Fc, fragment crystallisable.

**Figure 3 foods-10-01437-f003:**
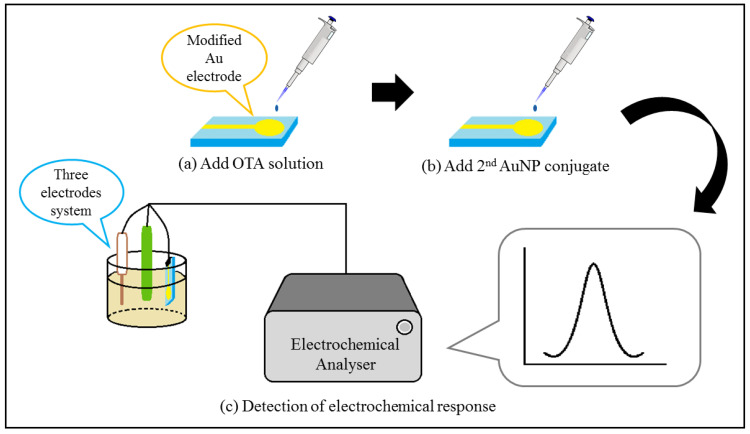
Schematic diagram illustrating the operation principle of indirect mycotoxin detection with a DPV aptasensor. (**a**) OTA solution was added onto the surface of modified Au electrode. (**b**) The signal amplifier, 2nd AuNP conjugate (ferrocene-tagged AuNPs) was added to compete with the OTA in the hybridisation with CP2. (**c**) The resultant electrochemical response was collected and analysed by the electrochemical analyser.

**Figure 4 foods-10-01437-f004:**
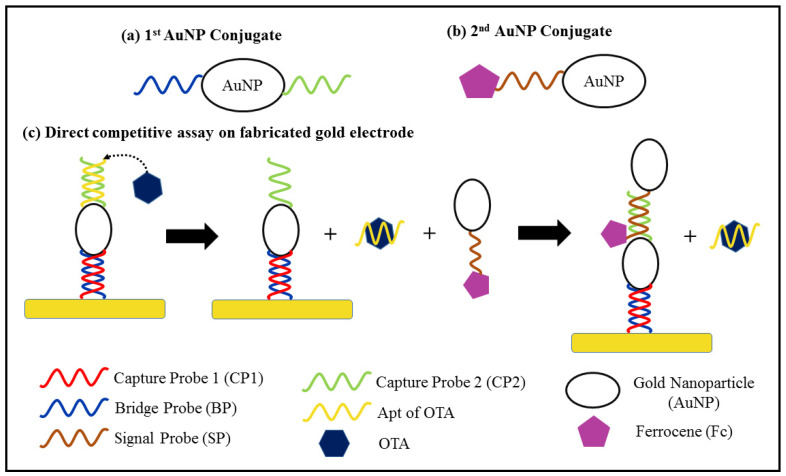
Schematic diagram of the (**a**) structure of the first AuNP conjugate (**b**) structure of the second AuNP conjugate and (**c**) indirect competitive detection of OTA.

**Figure 5 foods-10-01437-f005:**
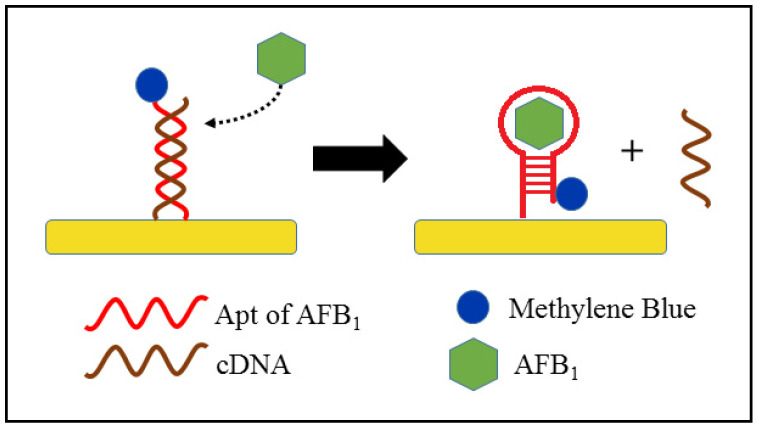
Schematic diagram of the AFB_1_ detection mechanism by a methylene blue-tagged aptasensor.

**Table 1 foods-10-01437-t001:** Application of antibodies for mycotoxin detection in food samples.

Immunoassay	Antibody	Mycotoxin	Sample	Half Maximal Inhibitory Concentration, IC_50_ (µg kg^−1^)	Limit of Detection, LOD (µg kg^−1^)	Linear Range (µg kg^−1^)	Percent Recovery (%)	Reference
Direct competitive ELISA	Broad-specific monoclonal antibody (mAb)	Total AFs (AFB_1_, B_2_, G_1_, G_2_)	Maize	0.04–0.06	0.21	0.001–0.81	74.5–96.5	[28]
FLISA	mAb	AFB_1_	Cereal	0.4	0.01	0.08–1.97	78.36–91.87	[27]
Immuno-chromatographic strip (ICS)	Single-chain variable fragment (scFv)	FUM B_1_	Maize	12.67	25	2.10–76.45	-	[32]

**Table 2 foods-10-01437-t002:** Detection of mycotoxins based on different instrumentation systems and sample pre-concentration.

Instrumentation (Phase System, Column)	Mobile Phase for Liquid Chromatography (LC) Column	Mycotoxin	Pre-Concentration Step	Response Time (min)	LOD (µg kg^−1^)	Linear Range (µg kg^−1^)	Percent Recovery (%)	Reference
UHPLC-MS/MS (Reverse phase, Acquity UPLC Ethylene Bridged Hybrid (BEH) C_18_ column)	Methanol (aq), 0.1% formic acid Methanol (aq) 0.5 mM ammonium acetate	AFB_1_ AFB_2_ AFG_1_ AFG_2_ AFM_1_	SPE (IAC)—AflaOcha HPLC	4	0.001–0.008	25–1 × 10^4^	80.0–110.0	[80]
HPLC-FLD (Reverse phase, Unimicro Technology C_18_ column)	Methanol 0.5% acetic acid (aq)	AFB_1_ AFB_2_ AFG_1_ AFG_2_ OTA	SPE (IAC)—AflaOcha HPLC	-	0.04 0.02 0.08 0.03 0.30	0.20–50.0 0.06–15.0 0.30–50.0 0.09–15.0 1.0–50.0	>62.0%	[77]
HPLC-MS/MS (Reverse phase, Hypersil GOLD C_18_ column)	0.05% formic acid (aq) Acetonitrile, 0.05% formic acid	AFB_1_ AFB_2_ AFG_1_ AFG_2_ OTA ZEN T-2	In-lab mIAC	-	0.10 0.04 0.10 0.04 0.20 0.10 0.40	0.30–25.0 0.12–20.0 0.30–20.0 0.12–20.0 0.60–30.0 0.30–25.0 1.2–40.0	98.8–102.3	[81]
HPLC-FLD (Reverse phase, Alltima C_18_ column)	2.0% acetic acid (aq) Acetonitrile	OTA	Apt-polyhedral oligometric silsesquioxane (POSS)-monolithic column	30	0.025	0.045–0.2	>92.2%	[84]
HPLC-FLD (Reverse phase, Alltima C_18_ column)	Acetonitrile, Tris-EDTA (TE) buffer	OTA	Poly(POSS-methacryl-co-N,N’-methylene-bisacrylamide-co-2-Acrylamido-2-methyl propane sulfonic acid-Apt (PMAA)-monolithic column	-	0.06	0.06–5.0	94.9–99.8	[85]
HPLC-DAD-FLD (Reverse phase, ZORBAX StableBond-C_18_ column)	Ultra-pure water Acetonitrile	DON	mIAC–Huan Magnech Bio-Tech	30	1.5–20.0	100–500	75.8–118.2	[78]
		3-Acetyldeoxynivalenol (3-AcDON)				100–500		
		15-Acetyldeoxynivalenol (15-AcDON)				100–500		
		ZEN				20–200		
		α-Zearalenol (α-ZOL)				20–200		
		β -Zearalenol (β-ZOL)				20–200		
		Zearalanone (ZAN)				20–200		
		α-Zearalanol (α-ZAL)				20–200		
		β-Zearalanol (β-ZAL)				20–200		
HPLC-MS/MS (Reverse phase, Zorbax Eclipse C_18_ column)	Water Methanol (aq), 5 mM ammonium acetate	AFB_1_ AFB_2_ AFG_1_ AFG_2_	No SPE required	9	0.16 0.11 0.36 0.16	0.225–1.25	50.0–120.0	[74]
HPLC-Photochemical Derivatisation (PCD)-FLD (Reverse phase, Agilent CAPCELL PAK-C_18_ column)	Water, methanol and acetonitrile (isocratic eluent)	AFB_1_ AFB_2_ AFG_1_ AFG_2_	SPE (IAC)–ToxinFast	-	0.4 0.5 0.4 0.3	0.625–50.0 0.156–12.5 0.625–50.0 0.156–12.5	74.5–88.2	[82]
HPLC-PCD-FLD (Reverse phase, Venusil MP C_18_ column)	Methanol and water (isocratic eluent)	AFB_1_	In-lab SPE (AAC)	12	0.05	-	91.8–108.6	[83]
HPLC-FLD (Reverse phase, Alltima C_18_ column)	2.0% Acetic acid (aq) Acetonitrile	OTA	Apt-MIP-monolithic column	-	0.05	0.14–1.0	95.5–105.9	[86]
UHPLC-MS/MS (Reversed phase, Shim-pack XR-ODS-III C_18_ column)	Water, acetonitrile	PAT	Solid-phase microextraction (SPME)	-	0.334	0.001–1.250	85.4–106.0	[45]

**Table 3 foods-10-01437-t003:** Detection of mycotoxins with different types of ELISA technique.

ELISA Technique	Signal Producer	Substance for Labelling the Competing Agent	Mycotoxin	Half Maximal Inhibitory Concentration (IC_50_)	LOD (µg kg^−1^)	Linear Range (µg kg^−1^)	Percent Recovery (%)	Reference
Colorimetric direct competitive	Bromocresol purple (BCP)	Glucose oxidase (GOx)	AFB1	0.066	-	0.025–0.2	82–115	[98]
Colorimetric direct competitive	Horseradish peroxidase (HRP)	Glucose oxidase (GOx)	AFB1	0.0223	0.004	0.0031–0.1500	80.56–108.53	[99]
Dynamic light scattering direct competitive	AuNP solution	Glucose oxidase (GOx)	AFB1	0.00136	0.00012	-	90.60–107	[100]
Direct competitive ULISA	Up-conversion nanoparticles (UNCP, type NaYF4:Yb,Tm) Streptavidin (SA)	Up-conversion nanoparticles (UNCP, type NaYF4:Yb,Tm) Streptavidin (SA)	ZEN	0.16 ± 0.08	0.02	-	77–105	[109]

**Table 4 foods-10-01437-t004:** Improvement of immunoassays in terms of molecular recognition elements on the basis of ELISA techniques.

Immunoassay	Molecular Recognition Element	Mycotoxin	Half Maximal Inhibitory Concentration, IC_50_ (µg kg^−1^)	LOD (µg kg^−1^)	Linear Range (µg kg^−1^)	Percent Recovery (%)	References
Direct ELISA	Monoclonal antibody	AFB_1_ Total AFs (AFB_1_, B_2_, G_1_, G_2_	0.037 ± 0.002 0.031 ± 0.001	0.38 0.43	-	97.1–107.3	[103]
Direct competitive ELISA	Nanobody Nb28	AFB_1_	0.75	0.13	0.24–2.21	84.2–116.2	[107]
Direct competitive ULISA	Peptide mimotope	ZEN	11	4.2	-	87–106	[108]
Competitive NAISA	Apt	AFB_1_	-	0.005	0.01–1000	80–105.2	[75]

**Table 5 foods-10-01437-t005:** Comparison of aptasensor performance by the EC techniques and the materials used at the electrode surface for signal amplification.

EC Technique	Types of Working Electrode	Surface Conductivity Enhancer	Supporting Substances/Signal Amplifier	Mycotoxin	Real Sample	LOD (µg kg^−1^)	Linear Range (µg kg^−1^)	References
EIS	Glassy carbon	AuNPs	-	PAT	Apple juice	0.046	0.154–1541.2	[113]
	Boron-doped diamond	AuNPs	-	AFB_1_	Peanut powder	1.718 × 10^−5^	3.123 × 10^−5^–3.123	[117]
	Glassy carbon	Poly(diallyl dimethylammonium chloride) graphene nanosheets Carboxylated polystyrene nanospheres	-	AFB_1_	Oil, soy sauce	0.002	0.001–0.1	[116]
	Glassy carbon	Platinum nanoparticles Metal–organic frameworks (MIL-101 (Fe))	-	AFM_1_	Milk powder, pasteurised milk	0.002	0.01–80	[119]
DPV	Au	AuNPs	AuNPs Ferrocene (Fc)	OTA	Wine	0.001	0.001–500	[125]
	Glassy carbon	Carboxylated graphene	-	OTA	Wine	1.333 × 10^−6^	4.038 × 10^−6^–4.038	[121]
	Indium-doped tin oxide (ITO) sheet	Carboxylated graphene	-	OTA	Grape juice	0.01	-	[120]
	Pencil graphite	-	-	ZEN	Cornflour, cornstarch, malt	29.47	100–600	[129]
	Au	Trogtalite (CoSe_2_) High crystallisation structure	Metal–organic frameworks (MOFs) Platinum-nickel (PtNi)	ZEN	Maize	1.37 × 10^−6^	1 × 10^−5^–10	[122]
	Glassy carbon	AuNPs	DNA-AuNPs-HRP Exonuclease	AFB_1_	Peanut, corn	3.3 × 10^−4^	0.001–200	[123]
	Au	-	Metal–organic framework Silver–platinum (AgPt) Iron–porphyrin (PCN-223-Fe)	OTA	Wine	1.4 × 10^−5^	2 × 10^−5^–2	[128]
	Glassy carbon	AuNPs Reduced molybdenum disulphide (rMoS_2_)	Gold nanoparticles Thionine (Thi) 6-(Ferrocenyl) hexanethiol (FC6S)	ZEN, FUM B_1_	Maize	5 × 10^−4^	0.001–10	[126]
	Au	Metal-organic frameworks (Fe-based) Gold-Platinum (Pt@AuNRs) Polyethyleneimine-reduced graphene oxide (PEI-rGO)	Nicking endonuclease (Nb.BbvCl)	PAT	Apple juice, apple wine	4.14 × 10^−5^	5 × 10^−5^–0.5	[127]
SWV	Au	-	Silver metallisation	OTA	Beer	7 × 10^−4^	0.001–100	[118]
	Au	-	Methylene blue	AFB_1_	Beer, white wine	0.625	0.625–1249	[130]
	Au	-	Methylene blue	AFB_1_	Wine, milk, cornflour	0.002	0.002–7.807 7.807–938	[56]

**Table 6 foods-10-01437-t006:** Summary of various commercially available mycotoxin detection kits.

Methods	Products	Time Required (min)	Mycotoxin	LOD (µg kg^−1^)	Quantification Range/Highest Limit (µg kg^−1^)	Qualitative/Quantitative	On-Site Detection	Manufacturer
Lateral Flow	AgraStrip	3	Total AFs (AFB_1_, B_2_, G_1_, G_2_)	3.3	0–500	Both	Yes	Romer Labs
		3	DON	250	250			
		3	FUM	150	250			
		3	ZEN	30	40			
		3	OTA	4	4			
	Reveal Q+	6	AFB_1_	2	3–100	Quantitative	Yes	
		3	DON	300	300–6000			
		6	FUM	300	300–6000			
		9	OTA	2	2–20			
		6	T-2 HT-2	50	50–600			
		5	AFM_1_	0.15	0.15–0.6			
	Reveal Q+ MAX	6	AFB_1_	3	3–50	Quantitative	Yes	
		5	T-2 HT-2	50	50–500			
		5	OTA	1.1	2–25			
		3	DON	300	300− 600			
		5	ZEN	21, 36	25–500			
	Smart Strip	5−10	AFB_1_	-	1–75	Both	Yes	Eurofins Technologies
		10	Total AFs (AFB_1_, B_2_, G_1_, G_2_)	-	2–75			
		10	DON	-	125–12,500			
		5	FUM	-	150–4000 750–20,000 (by dilution)			
		10	ZEN	-	50–1000 100–2000 (by dilution)			
	RIDA QUICK	5	Total AFs (AFB_1_, B_2_, G_1_, G_2_)	2	2–75 50–300	Quantitative	Yes	R-Biopharm
	QuickTox	2−4	Total AFs (AFB_1_, B_2_, G_1_, G_2_)	-	20	Qualitative	Yes	EnviroLogix
	QuickTox for QuickScan	5	Total AFs (AFB_1_, B_2_, G_1_, G_2_)	-	2.5–100	Quantitative	Yes	
		5	FUM	-	18,000			
		10	OTA	-	1.5–100			
		5	ZEN	-	50–520			
	TotalTox Comb	4	AFB_1_	-	2.7–30	Quantitative	Yes	
		4	DON	-	0.1–8			
		4	FUM	-	0.1–10			
		4	ZEN	-	50–500			
	ROSA AFQ-FAST	3−5	Total AFs (AFB_1_, B_2_, G_1_, G_2_)	-	5–30 20–100 50–300	Quantitative	Yes	Charm Sciences Inc.
	ROSA FAST5	5	DON	-	500–1500 1000–5400 >5000	Quantitative	Yes	
		5	FUM	-	500–1500 1000–5400 5000–25,000			
		5	ZEN	-	50–350 300–1000			
	ROSA DONQ2	2	DON	-	500–5400 400–30,000	Quantitative	Yes	
	ROSA AFQ-WETS5	5	Total AFs (AFB_1_, B_2_, G_1_, G_2_)	-	5–10 50–300	Quantitative	Yes	
	ROSA WET-S5	5	DON	-	500–5400 400–30,000	Quantitative	Yes	
		5	ZEN	-	50–1000			
	ROSA	10	T-2 HT-2	-	25–200 100–2000	Quantitative	Yes	
	Charm SLAFM	3	AFM_1_	0.35	-	Qualitative	Yes	
	Charm SLAFMQ	8	AFM_1_	0.5	-	Quantitative	Yes	
	Charm OCHRAQ-G	10	OTA	-	5–30 20–100	Quantitative	Yes	
	MycoTube	5	Total AFs (AFB_1_, B_2_, G_1_, G_2_)	>10	-	Qualitative	Yes	
	AflaSensor Quanti	10	AFM_1_	-	0.03–0.15	Quantitative	Yes	Unisensor
		10	AFM_1_	-	0.2–0.75			
	Rapid Test Strip	15	Total AFs (AFB_1_, B_2_, G_1_, G_2_)	5	-	Both	Yes	Nankai Biotech
			AFB_1_	5	-			
			ZEN	100	-			
			DON	500	-			
			OTA	50	-			
			T-2 HT-2	50	-			
			FUM	200	-			
ELISA	AgraQuant	15	Total AFs (AFB_1_, B_2_, G_1_, G_2_)	1–3	1–40	Quantitative	No	Romer Labs
		15	AFM_1_	0.0023–0.72	0.1–2			
		15	AFB_1_	2	2–50			
		15	OTA	1.9	2–40			
		15	ZEN	20	25–1000			
		15	FUM	200	250–5000			
		15	DON	200	250–5000			
		15	T-2	10	20–500			
	Agri-Screen	5	AFB_1_	20	-	Qualitative	Yes	Neogen
		10	DON	1000	-			
		15	FUM	5000	-			
	Veratox	5	AFB_1_	1.4	5–50	Quantitative	No	
		45	AFM_1_	0.0043	-			
		20	FUM	200	1000–6000			
		20	OTA	1	2–25			
		10	T-2 HT-2	25	25–250			
		10	ZEN	5	25–500			
	Veratox HS	20	AFB_1_	0.5	1–8	Quantitative	No	
		20	DON	25	25–250			
		15	FUM	50	50–600			
		30	OTA	1	2–10			
	Veratox HS	10	Total AFs (AFB_1_, B_2_, G_1_, G_2_)	2.5	5–50	Quantitative	No	
		15	ZEN	19.5	25–500			
	Celer	20	AFM_1_	0.025, 0.25	-	Quantitative	No	Eurofins Technologies
		15	Total AFs (AFB_1_, B_2_, G_1_, G_2_)	2	-			
		20	DON	40, 120, 240	-			
		20	FUM	750	-			
		20	OTA	2, 4	-			
		20	T-2	25	-			
		20	ZEN	10	-			
	B ZERO	15	AFB_1_	1	-	Quantitative	No	
		30	AFM_1_	0.01	-			
		20	DON	40, 120, 240	-			
		20	FUM	750	-			
		20	OTA	2, 4	-			
		20	T-2	25	-			
		20	ZEN	10	-			
	SENSISpec	10−20	Total AFs (AFB_1_, B_2_, G_1_, G_2_)	0.8–1.5	-	Quantitative	No	
	I’screen AFLA	50	Total AFs (AFB_1_, B_2_, G_1_, G_2_)	0.5, 1.25	-	Quantitative	No	
		75	AFM_1_	0.005, 0.05, 0.025, 0.037, 0.12	-			
	RIDASCREEN	45	OTA	0.5–1.6	0.3–30 0.6–60	Quantitative	No	R-Biopharm
	Screening Card	10	Total AFs (AFB_1_, B_2_, G_1_, G_2_)	Dependent on dilution	-	Qualitative	No	
		10	AFB_1_	Dependent on dilution	-			
		>15	OTA	<50	-			
	ELISA Kit	15	Total AFs (AFB_1_, B_2_, G_1_, G_2_)	1, 2	-	Quantitative	No	Biorex Food Diagnostics
		15	AFB_1_	1	-			
		20	ZEN	10	-			
		20	AFM_1_	0.025, 0.005	-			
		40	OTA	0.5, 1, 2	-			
	Plate Kit	20	Total AFs (AFB_1_, B_2_, G_1_, G_2_)	0.4, 0.6	1.2, 1.8	Quantitative	No	Beacon Analytical Systems
		75	AFM_1_	-	0.002–1			
		15	DON	-	200–2500			
		15	FUM	-	300–6000			
		15	ZEN	-	20–100			
		15	T-2 HT-2	-	25–500			
	Tube Kit	20	Total AFs (AFB_2_, G_1_, G_2_)	-	2–100	Quantitative	No	
		20	ZEN	-	10–100			
	ELISA Kit	25	AFM_1_	<0.005	0.005–0.135	Quantitative	No	Cusabio
		25	Total AFs (AFB_1_, B_2_, G_1_, G_2_, M_1_)	<0.02	0.02–1.62			
		25	AFB_1_	1, 2	0.15–4.05			
		25	OTA	<0.15	0.15–4.05			
		25	ZEN	<0.15	0.15–4.05			
		25	DON	<1	1–81			

**Table 7 foods-10-01437-t007:** Summary of current technologies with respect to their advantages and disadvantages for mycotoxin detection.

Mycotoxin Detection Technologies	Advantages	Disadvantages
HPLC	(1) High sensitivity (LOD as low as 0.001 µg kg^−1^) (2) Allow multi-detection of mycotoxin (3) Various choice of detectors available (4) Able to detect all common types of mycotoxin	(1) Expensive (2) Analyses easily interrupted by impurities (3) Require a pre-concentration step
GC-MS	-	(1) Require derivatisation (2) Complicated sample pre-treatment steps (3) Limited mycotoxin types are detected (4) Expensive
ELISA	(1) Qualitative and quantitative analyses are available (2) High-throughput screening with qualitative/quantitative analyses (80 samples at once) (3) High sensitivity (LOD as low as 0.0012 µg kg^−1^) (4) Nb28, mimotopes and Apts can overcome the disadvantages of an antibody	(1) High possibility of false positive/negative results in qualitative analysis (2) Long incubation period (> 2 to 3 h) (3) Antibody has low stability and flexibility, and low resistance to denaturation
EC Aptasensor	(1) Apts allow flexible modification of functional groups (2) High sensitivity (LOD as low as 1.333 × 10^−6^ µg kg^−1^) (3) Can detect a wide range of mycotoxin types due to its flexibility in electrode surface modification (4) Simple fabrication method (5) Simple operating procedure (6) Low cost of fabrication	(1) Requires surface modification and signal amplification to obtain high sensitivity
Lateral Flow	(1) Qualitative and quantitative analyses are available (2) Able to perform on-site detection (3) Simple operating procedure (4) Able to be coupled with a smartphone-based application	(1) Low sensitivity compared with other methods (LOD as low as 0.15 µg kg^−1^). (2) Antibody has low stability and flexibility, and low resistance to denaturation.

## Data Availability

No new data were created or analysed in this study. Data sharing is not applicable to this article.

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
