# Peer review of "Recent Advances in Conventional Methods and Electrochemical Aptasensors for Mycotoxin Detection"

_foods, 2021, doi:10.3390/foods10071437_

Round 1

Reviewer 1 Report

Dear editor, this review describes the conventional and electrochemical methods to detect mycotoxins in food and feed.

The manuscript was well organized and well elaborated. It covers a good amount of scientific literature, especially the recent one.

I can suggest inserting in the introduction an overview of the trend of publications in this field.

In addition, Section 2 should be renamed. Now it is named "material and methods," but I don't feel appropriate. 

Author Response

Response to Reviewer #1

Dear editor, this review describes the conventional and electrochemical methods to detect mycotoxins in food and feed. The manuscript was well organized and well elaborated. It covers a good amount of scientific literature, especially the recent one.

  1. I can suggest inserting in the introduction an overview of the trend of publications in this field.

ANSWER- Recently, the classification, toxicity, characteristic and contamination of mycotoxin in specific food and feeds are reviewed along with several analytical methods including instrumental and sensor techniques. (Ali, Hasan et al. 2020, Singh & Mehta 2020, Yang, Li et al. 2020). Biosensors with different recognition elements that focus on nanostructured materials and analytical techniques such as electrochemistry, electrochemiluminescence and photoelectrochemistry are the recent trend in mycotoxin [12,13,14]. The overview of the trend of publications in this field is added in the introduction. Correction has been done in page 2, line 56-62 of the revised manuscript.

  1. In addition, Section 2 should be renamed. Now it is named "material and methods," but I don't feel appropriate.

ANSWER- The title “Materials and Methods” has been changed to “Conventional and Advanced Analytical Technologies”. Correction has been done in page 1, line 72 of the revised manuscript.

Reviewer 2 Report

The review entitled “Recent Advances in Conventional Methods and Electrochemical Aptasensors for Mycotoxin Detection in Foodstuffs and Feedstuffs” includes the characteristics of various analytical methods used for the detection of mycotoxins in food and feed products.  The topic discussed in the paper is undoubtedly essential and actual, taking under consideration the chemical properties of fungal toxins as well as their toxicity. The paper is quite interesting, but in my opinion it should be a little remodel. The most important points are listed below:

  • The title should be shorter but informative;
  • The introduction part should be enriched in table including the most important mycotoxins contaminating food or feed products, with general chemical structures, fungal genera responsible for the production, and type of food/feed products;
  • The title of part 2 (Material and methods) should be changed; this title is usually used in experimental papers, in part containing an explanation and some details of the methodology. In this review, the title of part 2 should contain information about the real contents of this part – reading this title, you should know what you can find in this particular part;
  • The further part of the body text should be rearranged – first it is worth to discuss the conventional methods of mycotoxin detection and then describe the modern techniques with the discussion of the progress in the research considering different electrochemical transducers;
  • Additionally, it is worth to check every part and remove the basic almost handbook information (e.g., the antibody description in 2.1.1)
  • Finally, the conclusion part should be filled with the table summarizing all described methods with their advantages and disadvantages and with the information about which methods are used nowadays for commercial detection.

Author Response

Response to Reviewer #2

The review entitled “Recent Advances in Conventional Methods and Electrochemical Aptasensors for Mycotoxin Detection in Foodstuffs and Feedstuffs” includes the characteristics of various analytical methods used for the detection of mycotoxins in food and feed products.  The topic discussed in the paper is undoubtedly essential and actual, taking under consideration the chemical properties of fungal toxins as well as their toxicity. The paper is quite interesting, but in my opinion it should be a little remodel. The most important points are listed below:

  1. The title should be shorter but informative;

ANSWER- The title has been shortened to “Recent Advances in Conventional Methods and Electrochemical Aptasensors for Mycotoxin Detection”.

  1. The introduction part should be enriched in table including the most important mycotoxins contaminating food or feed products, with general chemical structures, fungal general responsible for the production, and type of food/feed products;

ANSWER- The most important mycotoxin contaminating food or feed products are mentioned in page 1, line 39-42. More information regarding this are added in page 1, line 42-44 and page 2, line 49. The general chemical structures are added as Fig. 1 in page 2, line 45-47 while its related text is added in page 1, line 36-38. The fungal general responsible for the production of mycotoxin is added in page 2, line 51-52.

  1. The title of part 2 (Material and methods) should be changed; this title is usually used in experimental papers, in part containing an explanation and some details of the methodology. In this review, the title of part 2 should contain information about the real contents of this part – reading this title, you should know what you can find in this particular part;

ANSWER- The title “Materials and Methods” has been changed to “Conventional and Advanced Analytical Technologies”. Correction has been done in page 2, line 72 of the revised manuscript.

  1. The further part of the body text should be rearranged – first it is worth to discuss the conventional methods of mycotoxin detection and then describe the modern techniques with the discussion of the progress in the research considering different electrochemical transducers;

ANSWER- The review is structured accordingly by firstly discuss about the conventional methods followed by the advanced method which is the electrochemical aptasensor. The parts for electrochemical aptasensor is not discussed based on the timeline for each emerging transducers because the comparison of the performance for aptasensor with different EC techniques is the focus regardless of which transducer was first being used.

  1. Additionally, it is worth to check every part and remove the basic almost handbook information (e.g., the antibody description in 2.1.1)

ANSWER- The basic information and general mechanism have been removed. Correction has been done in page 3, line 120-122, page 5, line 161, page 5, line 178-179, page 13, line 418-419, page 22 and line 647-649 of the revised manuscript.

  1. Finally, the conclusion part should be filled with the table summarizing all described methods with their advantages and disadvantages and with the information about which methods are used nowadays for commercial detection.

ANSWER- The advantages and disadvantages of all described methods are summarised in Table 7 in the conclusion part. Correction has been done in page 29, line 984-685, line 696-697, page 30, line 698 of the revised manuscript. The methods which are used nowadays are added in page 29, line 685-687 whereas the methods for commercial detection is added in page 29, line 692-694.

Reviewer 3 Report

The current review provides very comprehensive and informative comparisons of different technologies applied for Mycotoxin detection.  The manuscript was well prepared.

Author Response

Response to Reviewer #3

The current review provides very comprehensive and informative comparisons of different technologies applied for Mycotoxin detection.  The manuscript was well prepared.

ANSWER- Thank you.

Reviewer 4 Report

The review "Recent Advances in Conventional Methods and Electrochemical Aptasensors for Mycotoxin Detection in Foodstuffs and
Feedstuffs", written by Ong et al., describes set of methods applicable in the analysis of various mycotoxins. The depth of the review is relatively good and authors were able to find relevant data in existing publications and product materials and critically compare them. The review could be helpful to many researchers working in the field of food toxicity, mycotoxins and other related branches. I have few comments:

  1. The structure of the review can be improved. The authors spend a lot of effort describing various methods and approaches used in the analytical science and at the end of the chapter describe its application in the analysis of mycotoxins. The authors can reduce general statements and description of approaches.
  2. Description of aptasensors, antibodies, and MIPS (whole part 2.1) should be included in the description of the whole methods as these to make the review better readable and not to split related parts into more chapters. Description of the general mechanism of function of particular approaches could be reduces as this can be found elsewhere in related literature. It can be hypothesized that the reader already knows these approaches.
  3. Figures describing the basic operation modes of various approaches should be replaced with images describing concrete application of such approaches in the analysis of mycotoxins and results of such developed methods.

Author Response

Response to Reviewer #4

The review "Recent Advances in Conventional Methods and Electrochemical Aptasensors for Mycotoxin Detection in Foodstuffs and Feedstuffs", written by Ong et al., describes set of methods applicable in the analysis of various mycotoxins. The depth of the review is relatively good and authors were able to find relevant data in existing publications and product materials and critically compare them. The review could be helpful to many researchers working in the field of food toxicity, mycotoxins and other related branches. I have few comments:

  1. The structure of the review can be improved. The authors spend a lot of effort describing various methods and approaches used in the analytical science and at the end of the chapter describe its application in the analysis of mycotoxins. The authors can reduce general statements and description of approaches.

ANSWER- The general statements and description of approaches are reduced. Correction has been done in page 4, line 123, page 5, line 154,177, page 6, line 192, 231-232,236, page 7, line 253-254, page 13, line 418-419, page 15, line 532, page 22 and line 647-649 of the revised manuscript.

  1. Description of aptasensors, antibodies, and MIPS (whole part 2.1) should be included in the description of the whole methods as these to make the review better readable and not to split related parts into more chapters. Description of the general mechanism of function of particular approaches could be reduces as this can be found elsewhere in related literature. It can be hypothesized that the reader already knows these approaches.

ANSWER- The whole part of section 2.1 has been removed and included in the description of the whole methods. Correction has been done in page 3, line 114-122, page 7, line 249-254 of the revised manuscript. The basic information and general mechanism have been removed. Correction has been done in page 3, line 120-122, page 5, line 161, page 5, line 178-179, page 13, line 418-419, page 22 and line 647-649 of the revised manuscript.

  1. Figures describing the basic operation modes of various approaches should be replaced with images describing concrete application of such approaches in the analysis of mycotoxins and results of such developed methods.

ANSWER- The figures which depict the molecular design and mechanism is necessary to enable a better understanding of the reaction on the modified electrode surface during mycotoxin detection. Nevertheless, Fig. 3 is added in order to describe the concrete application of such approach. Correction has been done in page 16, line 575 and page 20, line 593-595 of the revised manuscript.

Round 2

Reviewer 2 Report

Authors corrected the text as recommended. The manuscript benefited greatly from this. 

Reviewer 4 Report

Authors adequately answered all my comments and the revised version was considerably improved, compared to the original version.